# WHEN DOES IN-CONTEXT LEARNING FALL SHORT AND WHY? A STUDY ON *Specification-Heavy* TASKS

## ABSTRACT

In-context learning (ICL) has become the default method for using large language models (LLMs), making the exploration of its limitations and understanding the underlying causes crucial. In this paper, we find that ICL falls short of handling *specification-heavy* tasks, which are tasks with complicated and extensive task specifications, requiring several hours for ordinary humans to master, such as traditional information extraction tasks. The performance of ICL on these tasks mostly cannot reach half of the state-of-the-art results. To explore the reasons behind this failure, we conduct comprehensive experiments on 18 specification-heavy tasks with various LLMs and identify three primary reasons: inability to specifically understand context, misalignment in task schema comprehension with humans, and inadequate long-text understanding ability. Furthermore, we demonstrate that through fine-tuning, LLMs can achieve decent performance on these tasks, indicating that the failure of ICL is not an inherent flaw of LLMs, but rather a drawback of existing alignment methods that renders LLMs incapable of handling complicated specification-heavy tasks via ICL. To substantiate this, we perform dedicated instruction tuning on LLMs for these tasks and observe a notable improvement. We hope the analyses in this paper could facilitate advancements in alignment methods enabling LLMs to meet more sophisticated human demands.

## 1 INTRODUCTION

Large language models (LLMs) have demonstrated exceptional language capabilities (Brown et al., 2020; OpenAI, 2023c; Anil et al., 2023). Due to their immense parameter scale, the predominant usage method of LLMs is in-context learning (ICL), i.e., LLMs implicitly learn how to handle a task with only the task instruction and a few demonstrations (Brown et al., 2020). Enhanced by alignment techniques such as instruction tuning (Wei et al., 2021; Chung et al., 2022; Iyer et al., 2022) and reinforcement learning from human feedback (Ouyang et al., 2022), the "LLM+ICL" paradigm is capable of serving extensive human needs, forming the foundation for many applications (Jiao et al., 2023; Gao et al., 2023b; Kocoń et al., 2023; Tan et al., 2023; Dao & Le, 2023). This brings the increasing importance of understanding the ability boundaries and limitations of LLM+ICL.

In this paper, we find that LLM+ICL falls short of handling *specification-heavy* tasks. Specification-heavy tasks refer to tasks with complex and extensive task specifications, often requiring ordinary humans to undergo substantial training time to master. As an example, Figure 1 illustrates a part of the ACE 2005 event detection (Walker et al., 2006) task specifications. Its full annotation guideline (Consortium, 2005) spans 77 pages. Even when we try to describe the essential task content with minimal language in our prompt design, the final prompt requires about 200 tokens. In our empirical study (§ 2), we collect 18 specification-heavy tasks from the range of conventional natural language understanding tasks and evaluate 6 competitive LLMs including GPT-4 (OpenAI, 2023c), Vicuna (Vicuna, 2023), FLAN-UL2 (Tay et al., 2022), etc. Experimental results demonstrate that the ICL performance of these LLMs often falls far below half of the previous state-of-the-art (SoTA) achieved by fine-tuned small-scale models.

To explore *why LLM+ICL falls short on specification-heavy tasks* (§ 3), we conduct intensive error analyses along with dedicated analytical experiments and identify three main failure reasons: (1) Inability to specifically understand context. Specification-heavy tasks often require a fine-grained understanding of given contexts to finish meticulous tasks, but LLMs lack specific understanding

**ACE 2005 Event Annotation Specifications**

① **Basic Concepts**: An Event is a specific occurrence involving participants. … We will not be tagging all Events, but only examples of a particular set of types and subtypes. Specifically, we will be interested in annotating LIFE, MOVEMENT, TRANSACTION, BUSINESS, CONFLICT, CONTACT, PERSONNEL and JUSTICE Events and among these a particular set of subtypes. The types and subtypes will be more thoroughly discussed in below. …

② **Annotating Event Triggers**: An Event's Trigger is the word (in its scope) that most clearly expresses its occurrence. In many cases, this will merely be the main verb in the part of the sentence (extent) that most directly describes the Event. Note that the following examples mark in bold only those triggers that are main verbs. …

③ **Annotating Complex Examples**: Most of the rules for identifying Event triggers discussed so far seem to work fairly well for the more simple examples. But the real challenge is to use these rules consistently for the complex cases as well. …

**…**

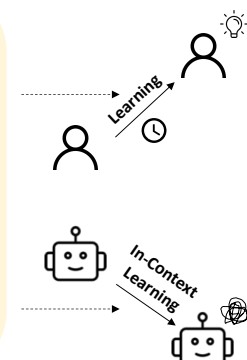

Figure 1: An example specification-heavy task: ACE 2005 event detection. This task involves heavy specifications which would take ordinary humans substantial time to learn. Therefore, solving the task using LLMs with in-context learning is challenging.

abilities. In the majority of error cases, LLMs either completely ignore the context, solely relying on their internal knowledge to make predictions (Longpre et al., 2021; Zhou et al., 2023b; Xie et al., 2023; Zhong et al., 2023) or overlook some important specific words within the context. (2) Misalignment in task schema comprehension with humans. Since the heavy task specifications often cannot be completely input to LLMs, specification-heavy tasks are often *underspecified* to LLMs. In the underspecified scenario, LLMs' understandings of task schema are often not fully aligned with human definitions (Si et al., 2023). As the task schema (event types) shown in Figure 1, we observe that LLMs consistently misclassify certain types, such as predicting BUSINESS as TRANSACTION. (3) Inadequate long-text understanding ability. It has been widely known that LLMs are often inadequate for understanding long contexts (Press et al., 2021; Shaham et al., 2022; Liu et al., 2023b). For specification-heavy tasks, this not only implies that LLMs would perform worse with longer given contexts similar to ordinary tasks, but more severely, we are unable to have LLMs fully utilize long task specifications, making the issue of underspecification hard to resolve.

For the aforementioned drawbacks of the LLM+ICL paradigm, *should we blame LLM or ICL*? To answer this question, we perform fine-tuning experiments to investigate the upper-bound performance of LLMs on the tasks (§ 4.1). Specifically, we fine-tune FLAN-UL2 (Tay et al., 2022), an LLM with 20 billion parameters, on each of the investigated specification-heavy tasks. The achieved results are much better than ICL performance and mostly comparable with existing SoTA. Moreover, we fine-tune a series of LLMs with different scales and observe a clear positive scaling effect, i.e., the fine-tuning performance on specification-heavy tasks improves with the increase in model size. These results indicate that the failure is not an inherent flaw of LLMs. The limitations on handling specification-heavy tasks come from ICL.

We posit that the inability of ICL to effectively handle specification-heavy tasks is due to the neglect of existing alignment methods. Existing alignment methods, such as instruction tuning (Wei et al., 2021; Chung et al., 2022) and RLHF (Ouyang et al., 2022), benefit from highly diverse data and tasks (Zhou et al., 2023a; Yue et al., 2023). However, existing alignment datasets often do not well cover complicated specification-heavy tasks (Wang et al., 2022d; Mishra et al., 2022b; Wang et al., 2022c; Wei et al., 2021; Longpre et al., 2023), resulting in the limitation of LLMs' ICL ability. To substantiate this, we conduct a preliminary experiment (§ 4.2). We perform straightforward instruction tuning to align FLAN-UL2 (Tay et al., 2022) on the investigated specification-heavy tasks. Following FLAN (Wei et al., 2021), we manually curate 10 instructions per task and diversify the set through augmentation techniques such as random shuffling of the predefined task schema. After our alignment, the ICL performance on specification-heavy tasks of FLAN-UL2 (20B) is obviously improved and reaches the level of text-davinci-003, which demonstrates the substantial potential of LLMs that could be unleashed with alignment. In light of this, we advocate for further research on alignment on specification-heavy tasks and discuss some possible directions (§ 5). Broadly, this will enable humans to fulfill more sophisticated demands with LLMs in the accessible ICL way.

| Type | Task | SoTA | FLAN-UL2 | Alpaca | Vicuna | ChatGPT | Davinci | GPT-4 |
|------|------|------|----------|--------|--------|---------|---------|-------|
| NER | CoNLL 2003 | 94.6 (Wang et al., 2021a) | 43.0 | 40.7 | 31.1 | 61.8 | 41.2 | 76.0 |
| | ACE 2005 | 89.5 (Zhang et al., 2022) | 4.7 | 15.9 | 24.6 | 34.0 | 32.8 | 42.3 |
| | FewNERD | 68.9 (Ding et al., 2021) | 1.8 | 18.1 | 17.0 | 44.1 | 31.2 | 52.2 |
| RE | TACRED | 76.8 (Wang et al., 2022a) | 2.9 | 0.0 | 0.0 | 7.3 | 15.8 | 25.2 |
| | SemEval | 91.9 (Cohen et al., 2020) | 14.0 | 9.2 | 6.2 | 24.0 | 16.1 | 39.5 |
| | FewRel 2.0 | 73.9 (Li et al., 2023b) | 10.0 | 0.0 | 0.0 | 46.0 | 40.0 | 68.0 |
| | DocRED | 67.5 (Ma et al., 2023) | 1.9 | 0.0 | 0.0 | 12.4 | 22.9 | 27.9 |
| ED | ACE 2005 | 73.5 (Wang et al., 2022a) | 0.5 | 3.5 | 4.3 | 27.0 | 22.6 | 33.7 |
| | MAVEN | 68.5 (Wang et al., 2021b) | 0.3 | 1.9 | 2.1 | 18.8 | 20.6 | 28.9 |
| | RichERE | 62.0 (Van Nguyen et al., 2022) | 0.0 | 5.1 | 1.7 | 18.8 | 15.3 | 23.8 |
| EAE | ACE 2005 | 72.7 (Ma et al., 2022) | 0.7 | 5.9 | 0.3 | 23.4 | 27.2 | 36.2 |
| | RichERE | 68.3 (Peng et al., 2023) | 0.2 | 10.6 | 6.3 | 28.7 | 29.2 | 41.0 |
| ERE | MATRES | 84.0 (Zhou et al., 2022) | 29.2 | 29.9 | 5.1 | 41.0 | 47.0 | 59.0 |
| | MAVEN-Causal | 31.5 (Wang et al., 2022b) | 1.4 | 17.6 | 1.0 | 16.3 | 9.0 | 9.0 |
| | MAVEN-Subevent | 27.5 (Wang et al., 2022b) | 5.2 | 6.7 | 15.4 | 24.8 | 1.5 | 2.2 |
| | MAVEN-Temporal | 56.0 (Wang et al., 2022b) | 12.1 | 6.8 | 6.9 | 13.2 | 30.4 | 31.3 |
| SA | GoEmotions | 46.0 (Demszky et al., 2020) | 29.6 | 18.3 | 11.9 | 27.4 | 26.7 | 31.8 |
| | SST-5 | 59.8 (Heinsen, 2022) | 45.3 | 31.1 | 39.2 | 55.0 | 54.0 | 58.0 |

Table 1: ICL performance (F1, %) of investigated LLMs on specification-heavy tasks.

## 2 PILOT EXPERIMENT: LLM+ICL FAILS ON SPECIFICATION-HEAVY TASKS

The section introduces pilot experiments on specification-heavy tasks, including investigated specification-heavy tasks (§ 2.1), experimental setup (§ 2.2), and experimental results (§ 2.3).

### 2.1 INVESTIGATED SPECIFICATION-HEAVY TASKS

Specification-heavy tasks involve complex specifications and typically require significant training time for ordinary humans to master. Based on the complexity of annotation guidelines, we collect 18 tasks across 6 different types from conventional natural language understanding tasks, including: (1) **Named Entity Recognition (NER)**, including CoNLL 2003 (Sang & De Meulder, 2003), ACE 2005 (Christopher et al., 2005), and FewNERD (Ding et al., 2021) tasks. The tasks aim to identify entities from texts and classify them into predefined types, such as *person*, *location*, etc. (2) **Relation Extraction (RE)**, including TACRED (Zhang et al., 2017), SemEval (Hendrickx et al., 2010), FewRel 2.0 (Gao et al., 2019), and DocRED (Yao et al., 2019) tasks. The tasks require extracting the relationship from a predefined relationship set between two entities mentioned in texts. (3) **Event Detection (ED)**, including MAVEN (Wang et al., 2020), ACE 2005 (Christopher et al., 2005), and RichERE (Song et al., 2015) tasks. The tasks aim to detect events from texts and classify them into predefined types, e.g., *attack*. (4) **Event Argument Extraction (EAE)**, including ACE 2005 (Christopher et al., 2005) and RichERE (Song et al., 2015). The tasks intend to extract arguments for events, e.g., *time*. (5) **Event Relation Extraction (ERE)**, including MATRES (Ning et al., 2018), aiming to extract temporal relations for events, and MAVEN-ERE (Wang et al., 2022b), which contains three tasks: MAVEN-Causal, MAVEN-SubEvent, and MAVEN-Temporal, aiming to extract causal, subevent, and temporal relations. (6) **Sentiment Classification (SC)**, including SST-5 (Socher et al., 2013) and GoEmotions (Demszky et al., 2020). The tasks require sentiment analysis of given texts and classifying them into an appropriate sentiment category, e.g., *positive*.

### 2.2 EXPERIMENTAL SETUP

We investigate several competitive LLMs, including **FLAN-UL2** (Tay et al., 2022), which is a FLAN-style (Wei et al., 2021) instruction tuned UL2 (Tay et al., 2022); **Alpaca** (Taori et al., 2023), which is aligned based on LLaMA (Touvron et al., 2023) using 52k high-quality instruction-following demonstrations; **Vicuna** (Vicuna, 2023), a LLaMA variant distilled from ChatGPT using 70K conversations; **GPT-3.5 Turbo** (OpenAI, 2022), abbreviated as **ChatGPT** in the following context; **text-davinci-003** (Ouyang et al., 2022), abbreviated as **Davinci** in the following context; **GPT-4** (OpenAI, 2023c). We conduct all experiments using human-written instructions and 8-shot demonstrations, except for 2-shot for DocRED and MAVEN-ERE (due to the limited input length), and 10-shot for FewRel 2.0 (to be consistent with previous SoTA). The demonstrations are sam-

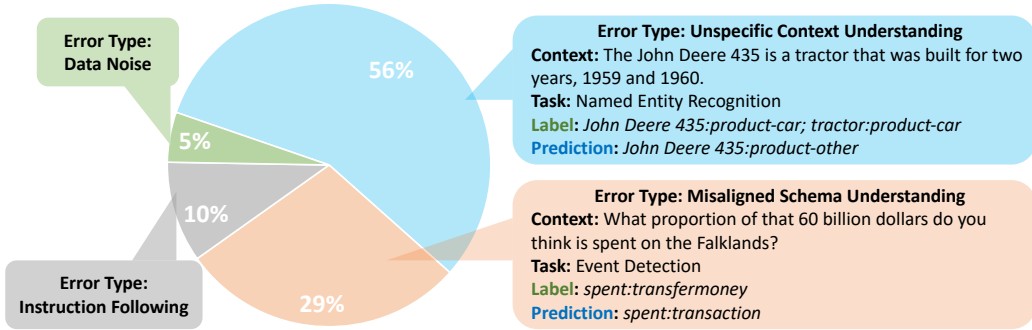

Figure 2: Error types with proportions from GPT-4. "*Unspecific Context Understanding*" means the lack of specific context understanding. "*Misaligned Schema Understanding*" represents LLMs' understanding of task schema is not fully aligned with humans. "*Instruction Following*" represents LLMs do not follow the instructions. "*Data Noise*" means the annotations are incorrect.

pled from the corresponding training set. Without loss of generality, we sample $1,000$ instances from each test set. If a test set contains fewer than $1,000$ instances, we incorporate the entire set. The evaluation metrics are all F1 scores calculated via string matching with ground-truth annotations (Liang et al., 2022). More experimental details are described in appendix A.1.

## 2.3 EXPERIMENTAL RESULTS

All the experimental results are shown in Table 1. We can observe that all LLMs perform poorly on the investigated specification-heavy tasks under ICL, especially the three open-sourced LLMs: FLAN-UL2, Alpaca, and Vicuna. Despite surpassing other LLMs, OpenAI's most advanced model, GPT-4, still often falls short half of previous SoTA models, almost all of which have less than 1B parameters. It indicates that specification-heavy tasks pose significant challenges to the existing LLM+ICL framework. We will explore why LLM+ICL fails on these tasks in the following sections.

## 3 WHY LLM+ICL FAILS?

To explore why LLM+ICL falls short on specification-heavy tasks, we conduct intensive error analyses based on the outputs of the top-performing GPT-4. Specifically, we sample $50$ error cases from FewNERD, TACRED, two ACE 2005 tasks, and three MAVEN-ERE tasks, respectively. We analyze and categorize four main error types, which are shown in Figure 2. We additionally conduct dedicated analytical experiments and identify three main failure reasons.

### 3.1 INABILITY TO SPECIFICALLY UNDERSTAND CONTEXTS

Specification-heavy tasks often require fine-grained comprehension of the information in given contexts to accomplish meticulous tasks, which is also why these tasks need extensive and detailed task specifications. However, we find that LLMs with ICL often lack fine-grained context understanding on these tasks, i.e., the inability to specifically understand context. As shown in Figure 2, around 56% of the errors can be attributed to unspecific context understanding. In these error cases, LLMs either ignore all the contexts and give predictions only based on their parametric knowledge (Longpre et al., 2021; Zhou et al., 2023b; Xie et al., 2023; Zhong et al., 2023) or overlook some important specific words within the contexts. As the example in Figure 2, LLMs neglect the word "*tractor*" in the context, which leads to the wrong type prediction for "*John Deere 435*".

We further conduct analytical experiments to validate the inability of LLMs to specifically understand contexts. We first sample a collection of $50$ instances from the *accurate* predictions on the FewNERD, TACRED, ACE 2005, MAVEN-ERE, and GoEmotions tasks. We then conduct minor modifications in the contexts of the sampled instances such as word replacement, resembling the text attack methods (Zang et al., 2020), and ensure the modifications change the golden labels. We evaluate GPT-4 on the modified instances and observe that more than half of predictions (27 out

of 50) remain unchanged. Among the unchanged instances, LLMs ignore all the contexts in 18 instances (67%) and utilize the contexts for predictions but neglect minor modifications in the other 9 instances (33%). It demonstrates that LLMs lack capabilities for specific context understanding. More details of analytical experiments are placed in appendix A.2.

## 3.2 MISALIGNMENT WITH HUMANS IN TASK SCHEMA COMPREHENSION

Specification-heavy tasks typically contain lengthy specifications, e.g., the specifications for ACE 2005 event detection span 77 pages (Consortium, 2005), hence it is nearly intractable to completely input them to LLMs via in-context learning. Therefore, specification-heavy tasks are inevitably *underspecified* for LLMs under the ICL setting. In the underspecified scenario, LLMs' understanding of tasks, e.g., task schema, may not be aligned with human expectations (Si et al., 2023). We find that the underspecification for specification-heavy tasks leads to a substantial proportion of errors.

| Proportion | Golden Label $\rightarrow$ False Prediction |
|---|---|
| 55% | Artifact $\rightarrow$ Agent |
| 82% | Chemical Thing $\rightarrow$ Medical |
| 87% | Transfer Money $\rightarrow$ Transaction |

Table 2: Cases of misclassification. The proportion refers to the number of a certain *incorrect* prediction divided by the number of positive instances for a golden label. LLMs consistently misclassify certain types to others.

As shown in Figure 2, about 29% of errors come from the misaligned schema understanding. For example, LLMs often confuse two event types `transaction` and `transfermoney` on ACE 2005 event detection. While human annotators can consult the specifications to understand the differences between the two types, the under-alignment of LLMs cannot be simply solved with ICL.

We further investigate the errors of GPT-4 and find it consistently misclassifies certain types. Table 2 shows several frequent instances where LLMs misclassify a type into another, for example, predicting `Chemical Thing` as `Medical`. It suggests that there surely exists a misalignment with humans in task schema comprehension. While eliminating the misalignment requires inputting extensive specifications to LLMs, we will point out in the next section that it is non-trivial due to the LLMs' inadequate long-text understanding ability.

## 3.3 INADEQUATE LONG-TEXT UNDERSTANDING ABILITY

It is widely known that LLMs lack sufficient capabilities to handle long texts (Press et al., 2021; Shaham et al., 2022; Liu et al., 2023b), which typically refer to the issue of modeling long contextual texts. We observe similar phenomena in specification-heavy tasks. As shown in Table 1, LLMs particularly underperform on tasks featuring long contexts: DocRED, MATRES, and MAVEN-ERE. We further investigate GPT-4 performance on DocRED instances with different context lengths and find that the performance consistently decreases as the contexts lengthen. Specifically, the F1 score decreases from 35% to 5% as the context length increases from 20 to 200 words. The full curve of the decreased performance is placed in appendix A.2.

The inadequate long-text understanding ability poses challenges to solving specification-heavy tasks with in-context learning, as specification-heavy tasks require extensive specifications to avoid underspecification issues. We further conduct experiments to explore whether extensive prompts can help LLMs solve the tasks. We sample 100 instances for five investigated tasks and employ more detailed descriptions of their task schemata rather than only minimal names. The results are shown in Table 3, and we can observe that utilizing extensive prompts even hurts the performance. To demonstrate more comprehensive trends, we also investigate the "*relative performance*", which is the ratio of LLM+ICL performance over SoTA indicating the difficulty of a task for LLM+ICL, on all the tasks with different specification lengths. Figure 3 shows that generally the longer the specification, the poorer the "*relative performance*" of the LLM+ICL paradigm[1], which further demonstrates the challenges that specification-heavy tasks present to the LLM+ICL paradigm. It suggests that due to inadequate long-text understanding ability, resolving the underspecification issue of specification-heavy tasks with ICL is difficult. More experimental details are shown in appendix A.2.

---

[1]There are some outliers (FewRel 2.0, FewNERD, and GoEmotions) in OpenAI's LLMs (ChatGPT, Davinci, and GPT-4), for which we postulate two potential reasons: (1) These LLMs may have been aligned on similar tasks. (2) The SoTA of FewRel 2.0 and FewNERD are relatively low due to original few-shot setting.

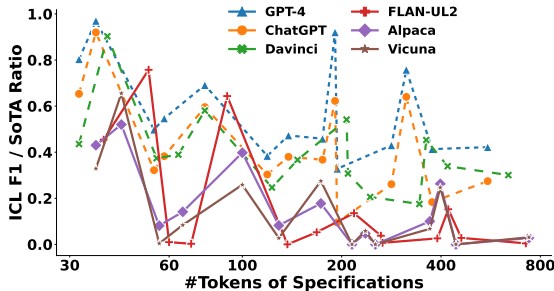

| Dataset | w/o desc. | w/ desc. |
|---|---|---|
| FewNERD | **48.9** | 48.5 |
| TACRED | **25.2** | 23.8 |
| ACE 2005 (ED) | 31.2 | **36.9** |
| ACE 2005 (EAE) | **39.5** | 39.3 |
| GoEmotions | **26.0** | 25.3 |

Figure 3: ICL F1 / SoTA ratios on all the tasks with specifications of varying length.

Table 3: ICL experimental results (%) of GPT-4 without (w/o) and with (w/) detailed descriptions (desc.) of task schema.

## 4 DO LLMs REALLY FAIL?

For the failures of LLM+ICL in specification-heavy tasks, *should we blame LLM or ICL*? Do LLMs inherently lack the ability to handle those tasks? If not, how to effectively handle specification-heavy tasks using LLMs? The section conducts comprehensive experiments to answer these questions.

### 4.1 FINE-TUNING LLMs ACHIEVES DECENT PERFORMANCE

The aforementioned analyses have showcased many issues of "LLM+ICL". To attribute the blame between LLMs and ICL, we unleash all the potentials of LLMs by fine-tuning them on specification-heavy tasks and observe their upper-bound performance. Specifically, we fine-tune FLAN-UL2 (Tay et al., 2022), an LLM with 20B parameters, on each of the investigated specification-heavy tasks.

**Experimental Setup** As FLAN-UL2 is a sequence-to-sequence model, we convert all the datasets into text-generation format. The input and output format for fine-tuning FLAN-UL2 is detailed in appendix A.3. Similar to Wang et al. (2022a), the output format is in the form of triplets separated with a special symbol. For each task, we fine-tune FLAN-UL2 on the training set and choose the model with the best performance on the validation set. We calculate the metrics via string matching, which is the same as in § 2.2. The hyper-parameters and other details are introduced in appendix A.3.

**Experimental Results** The experimental results are shown in Table 4. We can observe that fine-tuning FLAN-UL2 performs much better than in-context learning in Table 1. The fine-tuning results are on par with or even surpass previous SoTA. It demonstrates that existing LLMs are inherently capable of addressing specification-heavy tasks. Therefore, we *should not* attribute the failures of LLM+ICL on specification-heavy tasks to LLMs themselves.

**Scaling Law** We further investigate whether specification-heavy tasks can benefit from scaling up models. Specifically, we fine-tune FLAN-UL2 and the similar FLAN-T5 (Chung et al., 2022) model family (from SMALL to XXL). Figure 5 illustrates the curves of fine-tuning performance at different model scales. We present the average results of the same-type tasks. The detailed results of each task are shown in appendix A.3. We can observe a clear positive scaling effect, i.e., fine-tuned larger models perform better on specification-heavy tasks. It demonstrates that specification-heavy tasks do not possess particular characteristics for LLMs and the failures of LLM+ICL are mainly due to the limitations of in-context learning.

### 4.2 NEGLECT OF ALIGNMENT CAUSES ICL INABILITY

Why cannot in-context learning handle specification-heavy tasks? Previous studies have demonstrated that the strong generalization of ICL benefits from alignment on highly diverse data and tasks (Zhou et al., 2023a; Yue et al., 2023). However, tasks covered by existing alignment datasets usually can be specified in concise terms and not well cover complicated specification-heavy tasks (Wang et al., 2022d; Mishra et al., 2022b; Wang et al., 2022c; Wei et al., 2021; Longpre et al., 2023), which could limit LLMs' ICL capability and might also be the inherent reason for the fail-

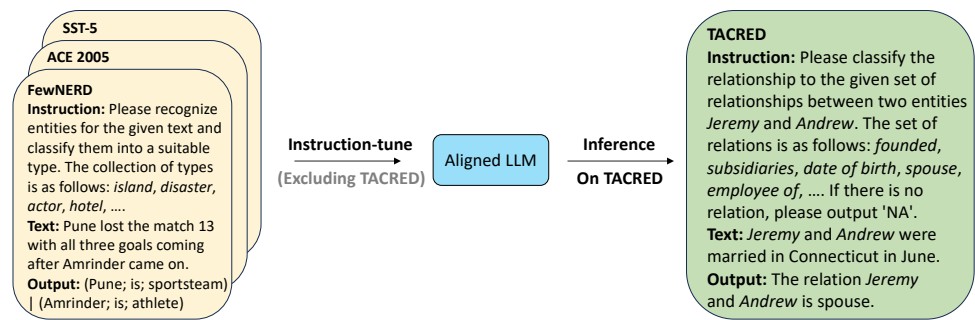

Figure 4: Illustration of instruction tuning. We hold out a task for evaluation and use the other tasks for instruction tuning. In this figure, we take TACRED as an example held-out task.

| Type | Task | FT | Aligned ICL |
|------|------|-----|-------------|
| NER | CoNLL 2003 | 92.5 | 52.3 |
| | ACE 2005 | 89.3 | 36.3 |
| | FewNERD | 67.4 | 38.7 |
| RE | TACRED | 72.7 | 12.7 |
| | SemEval | 87.9 | 16.1 |
| | FewRel 2.0 | 74.2 | 36.4 |
| | DocRED | 54.5 | 4.1 |
| ED | ACE 2005 | 70.5 | 29.0 |
| | MAVEN | 64.2 | 25.2 |
| | RichERE | 61.3 | 24.3 |
| EAE | ACE 2005 | 74.3 | 24.0 |
| | RichERE | 72.1 | 13.4 |
| ERE | MATRES | 35.9 | 57.6 |
| | MAVEN-Causal | 27.7 | 6.2 |
| | MAVEN-Subevent | 22.9 | 1.9 |
| | MAVEN-Temporal | 24.5 | 23.4 |
| SA | GoEmotions | 55.1 | 23.0 |
| | SST-5 | 62.0 | 38.8 |

Table 4: F1 scores (%) of FLAN-UL2 with fine-tuning (FT) and zero-shot ICL after our alignment (Aligned ICL).

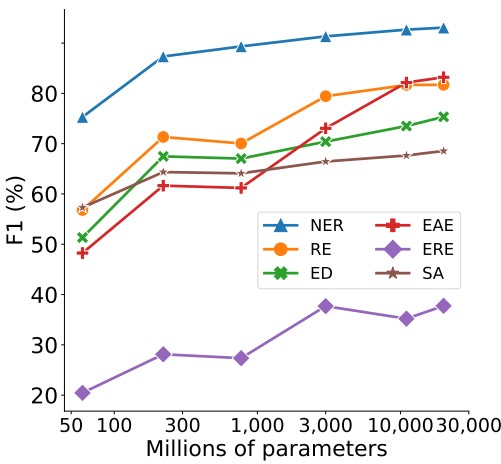

Figure 5: Scaling law of fine-tuning models at different scales on specification-heavy tasks. The reported results are the average within task types. Larger models perform better across all the investigated task types.

ures of ICL in specification-heavy tasks. To substantiate this assumption, we conduct preliminary experiments on aligning LLMs for specification-heavy tasks. Specifically, we align FLAN-UL2 with humans on specification-heavy tasks using a straightforward alignment method, instruction tuning (Wei et al., 2021; Chung et al., 2022).

**Experimental Setup** We construct the instructions of alignment data following the practice of FLAN (Wei et al., 2021). Specifically, we first manually curate 10 instructions for each task. The instructions consist of task descriptions and corresponding task schema. To diversify the instructions, we randomly shuffle the task schema and utilize various output formats, e.g., natural text outputs or triplets as in § 4.1. We use the training sets of considered tasks to instruction-tune FLAN-UL2. To validate the performance of our aligned model, we adopt the same evaluation method as FLAN, i.e., we hold out one task for evaluation while using all the other tasks for instruction-tuning. The instruction tuning and evaluation process is demonstrated in Figure 4. For evaluation, we adopt zero-shot evaluation, i.e., in-context learning with only instructions on test sets, which is also the same as FLAN. A more detailed experiment setup is shown in appendix A.4.

**Experimental Results** The experimental results are shown in Table 4. Compared with the results in Table 1, we can observe that after our instruction tuning, the zero-shot ICL performance of FLAN-UL2 is much better than the original few-shot ICL performance. The performance is even

comparable to ChatGPT and Davinci. It indicates that instruction tuning on specification-heavy tasks effectively aligns FLAN-UL2 with human expectations. After alignment on the tasks, FLAN-UL2 can well comprehend the basic instructions of the tasks and generalize to other tasks. Taking FewNERD as an example, this task contains 66 entity types. Directly handling FewNERD with underaligned FLAN-UL2 using in-context learning is difficult, resulting in an F1 of $1.8\%$. However, after alignment with instruction tuning, FLAN-UL2 significantly improves its performance to $38.7\%$ F1, while FewNERD is excluded from the training process and almost all types are unseen by LLMs. It reveals that current LLMs are underaligned on specification-heavy tasks, and the neglect of alignment causes the failures of in-context learning. In the LLMs era, we advocate for more research to enhance the alignment of LLMs with humans on specification-heavy tasks.

## 5 DISCUSSION

This section preliminarily discusses how to handle specification-heavy tasks and how to align LLMs with humans on specification-heavy tasks.

**Best Practice for Handling Specification-Heavy Tasks**  From a practical application perspective, fine-tuning models remain the most effective practice for handling specification-heavy tasks at present. As shown in Figure 5, the fine-tuning performance of FLAN-T5$_{BASE}$, which only has 250 million parameters, is significantly better than FLAN-UL2 (20B) with ICL, which has 80x more parameters. We also observe that continual training on instruction-tuned FLAN-UL2 for individual tasks can further enhance fine-tuning performance, which we place the details in appendix A.5. Fine-tuning performance on specification-heavy tasks consistently improves along with the increasing model size, but the computation cost is also higher, i.e., there is a trade-off between performance and computation cost. Therefore, one may adopt parameter-efficient fine-tuning (PEFT) in specificition-heavy tasks, which can achieve comparable performance to fine-tuning all parameters with lower cost (Houlsby et al., 2019; He et al., 2021; Ding et al., 2022). PEFT is also proven to be better and cheaper than in-context learning (Liu et al., 2022) and thus a competitive alternative.

In the era of LLMs, how to combine LLMs with fine-tuned small models is also an active area of research. One can enhance fine-tuning by using LLMs as tools, such as data augmentation using LLMs (Xu et al., 2023b; Whitehouse et al., 2023; Yu et al., 2023b). Many works have also explored the use of LLMs as agents in leveraging fine-tuned models as tools (Lu et al., 2023; Shen et al., 2023; Hsieh et al., 2023). Therefore, it is still worth exploring the combination of LLMs and fine-tuned models for specification-heavy tasks, which could potentially be a competitive practice in the future.

**Aligning LLMs with Humans on Specification-Heavy Tasks**  Alignment aims to align LLMs with human expectations (Wang et al., 2023; OpenAI, 2023b) and currently includes two main method categories: reinforcement learning from human feedback (RLHF) (Ouyang et al., 2022) and instruction tuning (Wei et al., 2021). In this paper, we preliminarily try to adopt instruction tuning to align LLMs on specification-heavy tasks. However, the aligned LLM still falls significantly short of the existing SoTA, indicating the need for further exploration of alignment methods.

Given the complexity of specification-heavy tasks, even humans need several rounds of trial and feedback to master these tasks. Inspired by this process, a possible alignment method is to decompose the task into multiple steps and align LLMs with humans step by step, which has been explored for mathematical reasoning (OpenAI, 2023a; Lightman et al., 2023). Taking the relation extraction task, which aims to extract the relationship between two entities, as an example, LLMs may be aligned in the following steps, which take inspiration from the human workflow of conducting relation extraction: (1) Let LLMs output the corresponding entity types of mentioned entities, which constrain the set of candidate relationships (Getoor & Taskar, 2007; Pawar et al., 2017). (2) Let LLMs determine the corresponding candidate relationships based on the entity types. (3) Evaluate one by one whether the two entities possess the relationship in the candidate set. The fine-grained alignment method may not only enhance performance on specification-heavy tasks but also improve the explainability of LLMs' output (Lightman et al., 2023).

Essentially, the major advantage of ICL is that it makes LLMs more accessible to average users without techniques such as fine-tuning. Alignment is the key to enhancing this (Zhou et al., 2023a; Wang et al., 2023). We believe that trying to handle specification-heavy tasks with ICL by better

aligning LLMs enhances LLMs' ability to cater to more complex and diverse human requirements, thus contributing to production and creativity development.

## 6 RELATED WORK

**Limitations of In-context Learning and LLMs**   One of the main limitations of in-context learning is its oversensitivity to many factors of prompt, including demonstration format (Mishra et al., 2022a), demonstration permutation (Lu et al., 2022; Zhao et al., 2021), and label word (Zhao et al., 2021), which poses a challenge for the application of ICL as poor prompts might even cause ICL falls into random guessing (Dong et al., 2022; Weng, 2023). Although ICL has become the default method for using LLMs, its specific working conditions are still unclear. Many studies find that in-context learning can still perform well even when using "unreasonable prompts" (Kaddour et al., 2023), such as irrelevant prompt (Webson & Pavlick, 2022), flipped or random labels (Min et al., 2022; Wei et al., 2023; Pan et al., 2023). These limitations may pose potential risks to the application of ICL (Ganguli et al., 2022; Perez et al., 2022).

In the era of LLMs, particularly since the introduction of ChatGPT (OpenAI, 2022), many works have focused on examining the limitations of LLMs' capabilities. Numerous studies have identified the limitations of LLMs in addressing certain natural language tasks, such as mathematical reasoning (Hendrycks et al., 2021; Bang et al., 2023; Frieder et al., 2023; Zhuang et al., 2023), logical reasoning (Liu et al., 2023a; Qin et al., 2023; Xu et al., 2023a; Bang et al., 2023), world knowledge recalling (Yu et al., 2023a; Sun et al., 2023a; Mallen et al., 2023), and information extraction (Jimenez Gutierrez et al., 2022; Li et al., 2023a; Han et al., 2023; Gao et al., 2023a). These works typically evaluate LLMs using in-context learning and often attribute failures to the limitations of LLMs themselves, overlooking the potential limitations of in-context learning. For example, Li et al. (2023a); Han et al. (2023) observe that LLMs underperform in information extraction (IE) tasks, which are mostly covered by specification-heavy tasks, and conclude that LLMs are incapable of effectively tackling IE tasks. However, through the decoupling analysis of LLMs and in-context learning, we find that it is the in-context learning, not the LLMs, that causes the poor performance.

In the paper, we identify the limitations of ICL in handling specification-heavy tasks and demonstrate that the neglect of alignment causes the limitations of ICL. We call for more research on uncovering the limitations of ICL for a more helpful, safe, and trustworthy application of LLM+ICL paradigm.

**LLM Alignment**   The general goal of alignment is to align AI models with **human expectations** (Kenton et al., 2021; Wang et al., 2023; OpenAI, 2023b), which is also the focus of alignment in the paper. The rise of LLMs raises broad safety and ethics concerns, rendering that recent alignment studies mainly focus on alignment with human values (Leike et al., 2018; Ray et al., 2019; Hendrycks et al., 2020; Gabriel, 2020; Tamkin et al., 2021; Bai et al., 2022a). The mainstream alignment methods can be primarily categorized into two types: reinforcement learning from human feedback (RLHF) (Ouyang et al., 2022; Bai et al., 2022b; Dong et al., 2023) and instruction tuning (Wei et al., 2021; Chung et al., 2022; Iyer et al., 2022; Wang et al., 2022c; Sun et al., 2023b; Zhou et al., 2023a; Li et al., 2023c). In the paper, we preliminarily utilize the instruction tuning method to align LLMs with humans on specification-heavy tasks. The performance of in-context learning is significantly improved after alignment, but still well below that of SoTA. We advocate for further research on developing more advanced alignment methods for specification-heavy tasks.

## 7 CONCLUSION AND FUTURE WORK

In this paper, we find that the dominating LLM+ICL paradigm falls short of handling specification-heavy tasks. We conduct intensive analyses and identify three main reasons for the failures: inability to specifically understand context, misalignment in task schema comprehension with humans, and inadequate long-text understanding ability. Furthermore, we discover that LLMs can handle specification-heavy tasks by fine-tuning, and these drawbacks come from the limitations of ICL. By preliminarily aligning an LLM on specification-heavy tasks with instruction tuning, we infer that the ICL inability is due to the neglect of existing alignment efforts. In the future, we will explore aligning LLMs on specification-heavy tasks using more advanced techniques such as the progress alignment method for mathematical reasoning (OpenAI, 2023a; Lightman et al., 2023).

## ETHICS STATEMENT

We discuss the ethical considerations and broader impact of this work here: (1) **Intellectual property**. Among our investigated tasks, the copyright of TACRED, ACE 2005, and RichERE belongs to LDC[2] and we access them through our LDC membership. All the other datasets are open-sourced, and we strictly adhere to their licenses. We believe all the datasets are well-desensitized. For the investigated LLMs, we query OpenAI's LLMs (ChatGPT, Davinci, and GPT-4) through paid APIs. For Alpaca and Vicuna, we strictly adhere to the `LLaMA` license[3], which originally is proposed for LLaMA (Touvron et al., 2023). We obtain the LLaMA's checkpoint by applying to Facebook[4]. (2) **Intended Use**. This paper finds the limitations of ICL in specification-heavy tasks and the reasons why ICL fails. We aim to provide meaningful insights regarding LLMs and ICL to the academic community through the intensive analyses in this paper, thereby promoting research on alignment in specification-heavy tasks. (3) **Misuse risks**. This paper reveals the inability of LLM+ICL in handling specification-heavy tasks. This inability could potentially result in erroneous or even harmful outputs. One **should not** exploit this flaw to attack LLMs for producing illegal information.

## REPRODUCIBILITY

To promote reproducibility, we provide experimental details in the appendices, including the details of pilot experiments (appendix A.1), analytical experiments (appendix A.2), fine-tuning (appendix A.3), and instruction tuning (appendix A.4). The evaluation source codes for the experiments are submitted as supplementary material.

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

APPENDICES

# A EXPERIMENTAL DETAILS

## A.1 PILOT EXPERIMENT

For ChatGPT, Davinci, and GPT-4, we use 8-shot demonstrations for all datasets except for 2-shot for DocRED and MAVEN-ERE (due to the limited input length), and 10-shot for FewRel 2.0 (the same setting as the previous few-shot learning). For the other LLMs, we can only use zero-shot demonstrations for DocRED and MAVEN-ERE due to the limited input length. For zero-shot experiments, we additionally incorporate an output format description in the prompt. For MATRES, MAVEN-Causal, MAVEN-Subevent, and MAVEN-Temporal, we require LLMs to output the relations for a single event pair at a time. Table 8 to 13 shows the input-output format of 6 representative tasks of each task type in the pilot experiments.

We query the APIs provided by OpenAI: `gpt-3.5-turbo`, `text-davinci-003`, and `gpt-4` for utilizing ChatGPT, Davinci, and GPT-4, respectively. The time period for our access to these APIs was from June 1 to June 30, 2023. All the API parameters are set to default. The FLAN-UL2, FLAN-T5, and Vicuna models are downloaded from Hugging Face Transformers (Wolf et al., 2020) and the model keys in Transformers are `google/flan-ul2`, `google/flan-t5-{size}`, and `lmsys/vicuna-7b-v1.1`, respectively. The Alpaca model is downloaded from its official GitHub repository[5]. We obtain the LLaMA's checkpoint by applying to Facebook[6]. The releases of Alpaca and Vicuna are the model diffs to LLaMA and we manually merge them. We use greedy search as the decoding strategy for all the open-sourced models: FLAN-UL2, FLAN-T5, Alpaca, and Vicuna.

## A.2 ANALYTICAL EXPERIMENTS

**Minor Modifications in Contexts**   We conduct minor modifications in the contexts of 50 instances sampled from the *accurate* predictions. An example of minor modification in contexts for relation extraction is shown in Table 14. We aim to minimize the number of modified words while ensuring the modification changes the gold label.

**Exploring whether LLMs Ignore all the Contexts**   To investigate whether LLMs ignore all the contexts or some specific words for the 27 instances with unchanged predictions (§ 3.1), we query GPT-4 with only the questions while excluding the contexts. Take relation extraction as an example, as shown in Table 15, we omit the contexts and directly query GPT-4 the relationships of two entities with only entity names: "*What is the relationship between Wen Qiang and bribes?*". We observe that in 18 cases (66.7%), LLMs predict the same with and without contexts, which suggests that LLMs ignore all the contexts and give predictions based on their parametric knowledge. For the other cases (33.3%), LLMs utilize the contexts for predictions but neglect specific words.

**Performance Degradation as Context Length Increases**   Figure 6 shows the full curve of performance degradation as context length increases on the DocRED task.

**Extensive Prompts with Schema Descriptions**   We employ detailed descriptions for predefined schemata in prompts. Specifically, we sample 100 instances each from FewNERD, TACRED, ACE 2005 (ED), ACE 2005 (EAE), and GoEmotions tasks for evaluation. Table 16 shows the schema descriptions for GoEmotions.

## A.3 FINE-TUNING

We fine-tune FLAN-T5 (Small, Base, Large, XL, and XXL) and FLAN-UL2 on investigated tasks as text generation tasks. The input and output formats of fine-tuning are the same as LLM+ICL in pilot experiments except that instructions and demonstrations are excluded from input. The hyperparameters for fine-tuning are shown in Table 6. The experimental results of all fine-tuned models on

---

[5] https://github.com/tatsu-lab/stanford_alpaca
[6] https://github.com/facebookresearch/llama

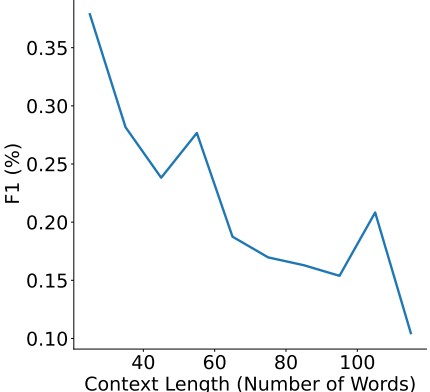

Figure 6: F1 scores (%) decrease as context length, i.e., the number of words, increases.

| FewNERD | TACRED | ACE 2005 (ED) | ACE 2005 (EAE) | GoEmotions |
|---|---|---|---|---|
| 67.8 | 73.4 | 74.0 | 75.8 | 52.9 |
| +0.4 | +0.7 | +3.5 | +1.5 | −2.2 |

Table 5: F1 scores (%) and improvements of continual fine-tuning on instruction-tuned FLAN-UL2. The improvements are compared to fine-tuning results in Table 4.

all the investigated specification-heavy tasks are shown in Table 7. All the fine-tuning experiments are conducted on Nvidia A100 GPUs, which approximately consume $1,000$ GPU hours.

## A.4 INSTRUCTION TUNING

We instruction-tune FLAN-UL2 on mixed datasets. The mixed datasets are sampled from original training sets and we sample up to 30k instances per dataset to balance the sizes of datasets. We utilize the examples-proportional mixing method (Raffel et al., 2020) in the training process and the mixing rate is 3k. We instruction-tune FLAN-UL2 with 32k gradient steps using AdamW (Loshchilov & Hutter, 2018) optimizer. The batch size is 32 and the learning rate is $1.0 \times 10^{-3}$. The maximum input and output sequence lengths are all $512$. The instruction tuning experiments are conducted on Nvidia A100 GPUs, which consume about $1,200$ GPU hours.

## A.5 CONTINUAL FINE-TUNING ON INSTRUCTION-TUNED FLAN-UL2

The effectiveness of fine-tuning could potentially be enhanced through existing techniques, such as two-stage fine-tuning (Phang et al., 2018; Li et al., 2019; Garg et al., 2020; Qiu et al., 2020). We follow this method to continue training on instruction-tuned FLAN-UL2 for individual tasks. We conduct experiments on FewNERD, TACRED, ACE 2005 (ED), ACE 2005 (EAE), and GoEmotions tasks and find that the results are consistently improved, which is shown in Table 5.

| | FLAN-T5$_{\text{SMALL}}$ | FLAN-T5$_{\text{BASE}}$ | FLAN-T5$_{\text{LARGE}}$ | FLAN-T5$_{\text{XL}}$ | FLAN-T5$_{\text{XXL}}$ | FLAN-UL2 |
|---|---|---|---|---|---|---|
| Epoch | 10 | 10 | 10 | 6 | 6 | 6 |
| Batch Size | 256 | 128 | 32 | 32 | 16 | 32 |
| Warmup Rate | 0.1 | 0.1 | 0.1 | 0.1 | 0.1 | 0.1 |
| Learning Rate | $3.0 \times 10^{-5}$ | $3.0 \times 10^{-5}$ | $1.0 \times 10^{-5}$ | $1.0 \times 10^{-5}$ | $5.0 \times 10^{-6}$ | $1.0 \times 10^{-5}$ |

Table 6: Hyper-parameters used in fine-tuning experiments.

| Task | FLAN-T5$_{SMALL}$ | FLAN-T5$_{BASE}$ | FLAN-T5$_{LARGE}$ | FLAN-T5$_{XL}$ | FLAN-T5$_{XXL}$ | FLAN-UL2 |
|---|---|---|---|---|---|---|
| CoNLL-2003 | 73.0 | 87.0 | 89.8 | 91.4 | 92.2 | 92.5 |
| ACE-2005 (NER) | 67.7 | 81.2 | 83.2 | 86.4 | 88.6 | 89.3 |
| FewNERD | 55.2 | 63.8 | 65.0 | 66.2 | 67.2 | 67.4 |
| TACRED | 64.2 | 70.0 | 70.3 | 73.0 | 73.2 | 72.7 |
| SemEval | 60.9 | 81.2 | 73.8 | 84.1 | 88.2 | 87.9 |
| FewRel 2.0 | 60.3 | 72.7 | 72.5 | 73.1 | 74.6 | 74.2 |
| DocRED | 15.3 | 32.8 | 36.0 | 51.2 | 53.6 | 54.5 |
| ACE 2005 (ED) | 40.0 | 62.2 | 61.3 | 64.9 | 69.0 | 70.5 |
| MAVEN | 48.7 | 60.0 | 60.4 | 62.3 | 64.4 | 64.2 |
| RichERE (ED) | 35.3 | 50.2 | 49.4 | 54.0 | 57.1 | 61.3 |
| ACE 2005 (EAE) | 33.9 | 42.5 | 42.4 | 59.4 | 72.8 | 74.3 |
| RichERE (EAE) | 42.6 | 60.8 | 60.0 | 66.7 | 71.5 | 72.1 |
| MATRES | 6.6 | 24.8 | 12.5 | 36.3 | 31.4 | 35.9 |
| MAVEN-Causal | 7.9 | 16.2 | 19.7 | 28.9 | 27.5 | 27.7 |
| MAVEN-SubEvent | 13.6 | 13.5 | 18.3 | 22.1 | 18.2 | 22.9 |
| MAVEN-Temporal | 13.7 | 18.0 | 18.9 | 23.5 | 23.7 | 24.5 |
| GoEmotions | 43.0 | 50.9 | 48.3 | 51.6 | 54.1 | 55.1 |
| SST-5 | 51.6 | 57.8 | 59.9 | 61.3 | 61.2 | 62.0 |

Table 7: Experimental results (%) of fine-tuned models on all the investigated tasks.

---

**Instruction**

Please recognize entities for the given text and classify them into a suitable type. The collection of types is as follows: art-broadcastprogram, art-film, art-music, art-other, art-painting, art-writtenart, building-airport, building-hospital, building-hotel, building-library, building-other, building-restaurant, building-sportsfacility, building-theater, event-attack/battle/war/militaryconflict, event-disaster, event-election, event-other, event-protest, event-sportsevent, location-bodiesofwater, location-GPE, location-island, location-mountain, location-other, location-park, location-road/railway/highway/transit, organization-company, organization-education, organization-government/governmentagency, organization-media/newspaper, organization-other, organization-politicalparty, organization-religion, organization-showorganization, organization-sportsleague, organization-sportsteam, other-astronomything, other-award, other-biologything, other-chemicalthing, other-currency, other-disease, other-educationaldegree, other-god, other-language, other-law, other-livingthing, other-medical, person-actor, person-artist/author, person-athlete, person-director, person-other, person-politician, person-scholar, person-soldier, product-airplane, product-car, product-food, product-game, product-other, product-ship, product-software, product-train, product-weapon.

---

**Demonstrations**

Text: Peter Nicol Russell's company, P. N. Russell and Company, constructed the heritage listed Denison Bridge at Bathurst.

Answer: Peter Nicol Russell: person-other; P. N. Russell and Company: organization-company; Denison Bridge: building-other; Bathurst: location-GPE;

`<other demonstrations >`

---

**Query**

Text: In 1926, before making his first-class debut, he played two matches for the Marylebone Cricket Club ( MCC ) against Ireland in Belfast and Dublin.

Answer:

---

**Answer**

*Marylebone Cricket Club: organization-sportsteam; MCC: organization-sportsteam; Belfast: location-GPE; Dublin: location-GPE;*

---

Table 8: An input-output example of FewNERD. The parts **Instruction**, **Demonstrations**, and **Query** are concatenated as the input to LLMs. **Answer** shows the output format.

---

**Instruction**

Please classify relationships between the two entities (marked with <entity> and </entity>). If the two entities have no relationships, please answer NA. Note the relation needs to be in the predefined set of relations. The set of relationships is as follows: org:founded, org:subsidiaries, per:date_of_birth, per:cause_of_death, per:age, per:stateorprovince_of_birth, per:countries_of_residence, per:country_of_birth, per:stateorprovinces_of_residence, org:website, per:cities_of_residence, per:parents, per:employee_of, per:city_of_birth, org:parents, org:political/religious_affiliation, per:schools_attended, per:country_of_death, per:children, org:top_members/employees, per:date_of_death, org:members, org:alternate_names, per:religion, org:member_of, org:city_of_headquarters, per:origin, org:shareholders, per:charges, per:title, org:number_of_employees/members, org:dissolved, org:country_of_headquarters, per:alternate_names, per:siblings, org:stateorprovince_of_headquarters, per:spouse, per:other_family, per:city_of_death, per:stateorprovince_of_death, org:founded_by.

---

**Demonstrations**

Text: Named for the chief theorist of modern Zionism, Theodor Herzl, <entity> Kollek </entity> was born in Nagyvaszony near Budapest in 1911 and raised in <entity> Vienna </entity>.
Answer: (Kollek; per:cities_of_residence; Vienna)
<other demonstrations>

---

**Query**

Text: This news comes from Karr Ingham, an economist who created the <entity> Texas Petro Index </entity> (TPI), which is a service of the <entity> Texas Alliance of Energy Producers </entity>.
Answer:

---

**Answer**

*(Texas Petro Index; org:parents; Texas Alliance of Energy Producers)*

---

Table 9: An input-output example of TACRED.

---

**Instruction**

Please identify the events in the text and classify them into appropriate categories; The collection of categories is as follows: Becoming, Creating, Process_start, Name_conferral, Presence, Departing, Recording, Reporting, Cause_to_make_progress, Bodily_harm, Arranging, Supply, Getting, Change, Hold, Come_together, Destroying, Hostile_encounter, Cause_change_of_position_on_a_scale, Conquering, Expressing_publicly, Killing, Dispersal, Agree_or_refuse_to_act, Coming_to_be, Communication, Bringing, Achieve, Sending, Change_event_time, Competition, Catastrophe, Attack, Surrounding, Military_operation, Change_sentiment, Award, Response, Commitment, Control, Process_end, Motion, Deciding, Change_of_leadership, Earnings_and_losses, Choosing, Sign_agreement, Placing, Death, Besieging, Escaping, Motion_directional, Receiving, Self_motion, Cause_to_amalgamate, Defending, Cause_to_be_included, Removing, Statement, Preventing_or_letting, Openness, Causation, GiveUp, Expend_resource, Aiming, Education_teaching

---

**Demonstrations**

Text: The 2008 event also included a MySpace bus, where secret sets and interviews with bands took place.
Answer: included: Cause_to_be_included; took place: Process_start
<other demonstrations>

---

**Query**

Text: The ruling National Command of the Arab Socialist Ba'ath Party were removed from power by a union of the party's Military Committee and the Regional Command, under the leadership of Salah Jadid.
Answer:

---

**Answer**

*removed: Removing*

---

Table 10: An input-output example of MAVEN.

**Instruction**
Please extract event arguments and their roles for the events marked with <event> and </event> in the text, the possible roles must be chosen from the Roleset. If there are no roles for the event, place output "NA".

**Demonstrations**
Text: Schrenko is a scumbag and, although she won't get any actual jail time, I hope the fine she pleads to is at least as much as what she <event> stole </event>.
Role Set: Beneficiary, Time-Within, Money, Time-Starting, Time-Before, Place, Recipient, Giver, Time-Holds
Answer: Schrenko: Recipient
`<other demonstrations>`

**Query**
Text: Earlier documents in the case have included embarrassing details about perks Welch received as part of his <event> retirement </event> package from GE at a time when corporate scandals were sparking outrage.
Role Set: Time-Within, Time-Starting, Time-After, Time-Before, Time-Ending, Place, Entity, Position, Person, Time-Holds
Answer:

**Answer**
*GE: Entity; Welch: Person*

Table 11: An input-output example of the ACE 2005 event argument extraction task.

**Instruction**

Please classify the relation between two events/Timex in a given document. There are 10 types of relations: [before, overlap, contains, simultaneous, begins-on, ends-on, cause, precondition, subevent, and coreference]. In each document, 2 events/Timex are marked as <event> event_name </event> or <Timex> Timex_name </Timex>. If there is a relation type or multiple relation types, the answer form is Answer: [relation type 1, relation type 2,...].

**Demonstrations**

Text: The 2012 Great Britain and Ireland floods were a series of weather events that affected parts of Great Britain and Ireland periodically during the course of 2012 and on through the winter into 2013. The beginning of 2012 saw much of the United Kingdom experiencing droughts and a heat wave in March. A series of low pressure systems steered by the jet stream brought the wettest April in 100 years, and <event> flooding </event> across Britain and Ireland. Continuing through May and leading to the wettest beginning to June in 150 years, with flooding and extreme events occurring periodically throughout Britain and parts of Atlantic Europe. On 27 and 28 June and again on 7 July heavy rain events occurred from powerful thunderstorms that gathered strength as they travelled across mainland Britain. Severe weather warnings and a number of flood alerts were issued by the UK's Environment Agency, and many areas were hit by flash floods that overwhelmed properties and caused power cuts. A motorist was killed after his vehicle was caught by floodwater and landslides halted rail services between England and Scotland. The thunderstorms were the product of two fronts that collided over the British Isles Ž2013 warm air travelling from the Azores and cold water-ladened air from the west. The second batch of flooding struck the South-West of England during the afternoon of 7 July, forcing the Met Office to issue its highest alert, Red (Take Action), due to the significant amounts of rainfall caused by a system travelling from Southern Europe, along with the warm, humid air the United Kingdom had seen in the run-up to the floods, which, like the June floods, caused thunderstorms. During the Autumn the most intense September low since 1981 brought widespread flooding and wind damage to the UK. Widespread flooding occurred again in November, December and <Timex> January 2013 </Timex> as more heavy rains overwhelmed the saturated ground.
The first event/ Timex : <event> flooding </event>
The second event/ Timex : <Timex> January 2013 </Timex>
Answer: [before]
<other demonstrations>

**Query**

Text: Tropical stormrumbia, known in the philippines as tropical storm unk, brought deadly flooding to the central and southern philippines in november and december 2000. The last of three consecutive tropical cyclones of at least tropical storm intensity to strike the Philippines, Rumbia <Event> began </Event> as a tropical depression on November 27, gradually intensifying to reach tropical storm intensity the next day. Strengthening later stagnated, and Rumbia would weaken back to depression status as it made landfall on the central Philippines on November 30. though the japan meteorological agency determined rumbia to have <Event> dissipated </Event> on december 2, the joint typhoon warning center continued to monitor the system over the next few days as it tracked across the south china sea. For a period of time beginning on December 5, Rumbia reorganized and strengthened back to tropical storm intensity before wind shear began to weaken the system. Located south of Vietnam on December 7, the storm's circulation center became devoid of convection, and by then Rumbia was declared by the JTWC to have dissipated. In the Philippines, Rumbia caused roughly US $1 million in damage and 48 fatalities. Several transportation routes were suspended in the lead-up to the storm's landfall. As a result of the tropical storm, power outages occurred, especially in Surigao. Several towns and villages were flooding, displacing around 70,000 people and putting 4,100 people into temporary emergency sheltering.
The first event/ Timex : <Event> began </Event>
The second event/ Timex : <Event> dissipated </Event>
Answer:

**Answer**

[*before*]

Table 12: An input-output example of MAVEN-ERE.

---
**Instruction**
Please comprehend the emotions expressed in the given text. The set of emotions is as follows: admiration, amusement, anger, annoyance, approval, caring, confusion, curiosity, desire, disappointment, disapproval, disgust, embarrassment, excitement, fear, gratitude, grief, joy, love, nervousness, optimism, pride, realization, relief, remorse, sadness, surprise, neutral.

---
**Demonstrations**
Text: This is amazing man. Congrats and keep em coming
Answer: admiration; gratitude
`<other demonstrations>`

---
**Query**
Text: Help, help, I'm being repressed!
Answer:

---
**Answer**
*fear*

---

Table 13: An input-output example of GoEmotions.

---
**Original**
Text: Judy Gross' husband <entity> Alan Gross </entity> was arrested at the <entity> Havana </entity> airport in December 2009.
Answer: (Alan Gross; per:cities_of_residence; Havana)

---
**Modified**
Text: Judy Gross' husband <entity> Alan Gross </entity> arrived at the <entity> Havana </entity> airport in December 2009.
Answer: NA
Model output: (Alan Gross; per:cities_of_residence; Havana)

---

Table 14: An example of minor modifications in contexts for relation extraction. In this example, we change the phrase "was arrested" to "arrived", and the golden label is also changed to NA.

---
**Instruction**
Please classify relationships between the two entities in the input. The input format is: (entity1; entity2). Answer the relation type only. You should classify the relationships based on your knowledge about the entities. If the two entities have no relationships, please answer NA. The set of relationships is as follows: org:founded, org:subsidiaries, per:date_of_birth, per:cause_of_death, per:age, per:stateorprovince_of_birth, per:countries_of_residence, per:country_of_birth, per:stateorprovinces_of_residence, org:website, per:cities_of_residence, per:parents, per:employee_of, per:city_of_birth, org:parents, org:political/religious_affiliation, per:schools_attended, per:country_of_death, per:children, org:top_members/employees, per:date_of_death, org:members, org:alternate_names, per:religion, org:member_of, org:city_of_headquarters, per:origin, org:shareholders, per:charges, per:title, org:number_of_employees/members, org:dissolved, org:country_of_headquarters, per:alternate_names, per:siblings, org:stateorprovince_of_headquarters, per:spouse, per:other_family, per:city_of_death, per:stateorprovince_of_death, org:founded_by.

---
**Query**
What is the relationship between Wen Qiang and bribes?

---
**Answer**
*per:charges*

---
**Model Prediction**
*per:charges*

---

Table 15: An example of TACRED omitting all the contexts. We evaluate LLMs using **Instruction** and **Query**, which contains only questions.

**Instruction**

Please comprehend the emotions expressed in the given text. The set of emotions is as follows: admiration, amusement, anger, annoyance, approval, caring, confusion, curiosity, desire, disappointment, disapproval, disgust, embarrassment, excitement, fear, gratitude, grief, joy, love, nervousness, optimism, pride, realization, relief, remorse, sadness, surprise, neutral. You should make the decision based on the definition of each type which is given as follows.

*admiration*: Finding something impressive or worthy of respect.
*amusement*: Finding something funny or being entertained.
*anger*: A strong feeling of displeasure or antagonism.
*annoyance*: Mild anger, and irritation.
*approval*: Having or expressing a favorable opinion.
*caring*: Displaying kindness and concern for others.
*confusion*: Lack of understanding, uncertainty.
*curiosity*: A strong desire to know or learn something.
*desire*: A strong feeling of wanting something or wishing for something to happen.
*disappointment*: Sadness or displeasure caused by the nonfulfillment of one's hopes or expectations.
*disapproval*: Having or expressing an unfavorable opinion.
*disgust*: Revulsion or strong disapproval aroused by something unpleasant or offensive.
*embarrassment*: Self-consciousness, shame, or awkwardness.
*excitement*: Feeling of great enthusiasm and eagerness.
*fear*: Being afraid or worried.
*gratitude*: A feeling of thankfulness and appreciation.
*grief*: Intense sorrow, especially caused by someone's death.
*joy*: A feeling of pleasure and happiness.
*love*: A strong positive emotion of regard and affection.
*nervousness*: Apprehension, worry, anxiety.
*optimism*: Hopefulness and confidence about the future or the success of something.
*pride*: Pleasure or satisfaction due to one's own achievements or the achievements of those with whom one is closely associated.
*realization*: Becoming aware of something.
*relief*: Reassurance and relaxation following release from anxiety or distress.
*remorse*: Regret or guilty feeling.
*sadness*: Emotional pain, sorrow.
*surprise*: Feeling astonished, startled by something unexpected.
*neutral*: Neither positive, negative, nor others.

**Demonstrations**

Text: This is amazing man. Congrats and keep em coming
Answer: admiration; gratitude
`<other demonstrations>`

**Query**

Text: Help, help, I'm being repressed!
Answer:

**Answer**

*fear*

Table 16: Detailed descriptions of each emotion option in the GoEmotions task.

