# OpenReview forum: "When does In-context Learning Fall Short and Why? A Study on Specification-Heavy Tasks"
_ICLR.cc/2024/Conference — Submitted to ICLR 2024_

### Official Review · Reviewer_hYr5 · 2023-10-25

**Soundness:** 2 fair
**Presentation:** 4 excellent
**Contribution:** 2 fair
**Rating:** 5
**Confidence:** 4

**Summary:**

This paper conducts a comprehensive evaluation of in-context learning and points out the inability of large language models in solving specification-heavy tasks.

To further trace the root of this issue, the authors also explore the performances of instruction-tuned models and fine-tune models with related data. Evaluation results indicate limitations are primarily due to the alignment data used for model training, rather than the large language models per se.

**Strengths:**

1. Engaging topic and comprehensive evaluation.

2. Meaningful discussions and exploration of fine-tuning.

3. The paper is very well-written and easy to follow, making it a pleasure to read.

**Weaknesses:**

1. One major weakness is the selection of datasets. Most of these focus on information extraction (IE), and as pointed out in previous work [1], IE tasks are not generally considered in most instruction-tuning datasets. Therefore, IE tasks are not only specification-heavy but also unfamiliar to these instruction-tuned models. A simple investigation/baseline should involve manually curating specification-heavy but familiar instructions such as:

The output length should not exceed X length, the output should start with…, the output should be in a JSON format, among other considerations.

These specifications are likely to arise in user cases or training data and are also specification-heavy. This provides a further disentanglement of the root of this inability. Also, ,ore interesting discussions may occur in this direction. For example, can general specifications assist complex task-specific specification understanding? And vice versa?

[1] Zhang et al, Aligning Instruction Tasks Unlocks Large Language Models as Zero-Shot Relation Extractors. Findings of ACL 2023.


2. Minor: Deeper analysis and insights are expected to see. Sec. 3.1 3.2 & 3.3 mainly explain the superfacial observations and these parts take large portion of the papers, I would prefer to see more analysis earlier in the paper.

**Questions:**

No

---

> ### Author Response · Authors · 2023-11-21
>
> Thanks for your detailed and insightful review!
>
> Regarding the Weakness #1: Thank you for the insightful suggestion. We will design tasks that are 'specification-heavy but familiar (to LLMs)' to analyze the sources of inability more effectively. Additionally, regarding the discussion on 'can general specifications assist complex task-specific specification understanding?', we intend to explore this in our future work. Our aim is to enhance LLMs' capabilities in instruction understanding and reasoning through the use of general specifications.
>
> Regarding the Weakness #2: In Section 3, we primarily focused on observing and experimentally analyzing errors in LLMs, aiming to provide insights for future work. Following your suggestion, we will condense this section in our subsequent revisions and include more extensive analysis at the modeling level.

---

> > ### Comment · Reviewer_hYr5 · 2023-11-22
> >
> > Thanks for your reply. Looking forward to see your update!

---

### Official Review · Reviewer_Huq3 · 2023-10-31

**Soundness:** 3 good
**Presentation:** 3 good
**Contribution:** 3 good
**Rating:** 5
**Confidence:** 3

**Summary:**

Noticing that the performance of ICL on specification-heavy tasks, specifically traditional IE tasks (such as named entity extraction and relation extraction, 18 tasks in total) mostly cannot reach half of the state-of-the-art results, this paper conducts preliminary error analysis and identifies three primary reasons that ICL falls short of handling these tasks. Moreover, this paper demonstrates that through fine-tuning and instruction tuning, LLMs can achieve decent performance on these tasks. This indicates that LLMs can perform specification-heavy tasks well if they are well aligned with human expectations.

**Strengths:**

1. This paper focuses on an important research question that ICL falls short of handling specification-heavy tasks and presents an explorative analysis. This paper will be of interest to practitioners who aim to prompt LLMs to complete complicated tasks via ICL.
2. The experiments on instruction tuning provide valuable findings, which properly demonstrate that well-aligned LLMs can perform unseen specification-heavy tasks better via ICL.
3. The presentation of this paper is clear, with few typos.

**Weaknesses:**

1. Though the motivation of this paper is impressive, the analysis underwhelms in some aspects.  Firstly, in Figure 2, how do the authors attribute the errors made by LLMs to the three types of reasons? Is there a chain-of-thoughts or any monologues to help identify the reasons? It seems to me that the example presented in "unspecific context understanding" can also be attributed to “Misaligned Schema Understanding”.  One possible explanation is that the model noticed the “tackor“ but didn’t treat it as “product-car“, because the task instruction missed important information about the class “product-car“. Secondly, some important experimental details are missing. What are the prompts of Figure 3 for different models? What is the "specification" in a prompt and how does its length vary? Besides, does the performance drop come from the longer context or the content of the specification?
2. What is the scope of “specification-heavy tasks”? This paper primarily focuses on traditional IE tasks (16 out of 18 tasks). However, there is no clear boundary of “specification-heavy" tasks in this paper.  In my opinion, most of NLP tasks involve “complex” annotation guidelines if we would like to elaborate detailed task descriptions and edge cases. Could the authors provide a clear definition of "specification-heavy task" or at least some examples of opposite tasks? It's also possible to narrow the scope of your paper to IE tasks, without losing generality.
3. The instructions for different tasks are too simple. At least the definition of different entity/relation types should be provided.

**Questions:**

1. After the instruction tuning, the zero-shot performance gain of different tasks varies greatly. Could the authors provide some analysis or intuition on it?
2. It would be intriguing to see to what degree the instruction tuning on specification-heavy tasks addresses three identified challenges.
3. This paper [1] is highly relevant to your second reason (misalignment in task schema comprehension with humans).
4. Typo - Figure 6 (remove %)

[1] Guideline Learning for In-context Information Extraction. EMNLP 2023.

---

> ### Author Response · Authors · 2023-11-21
>
> Thank you for your valuable review!
>
> Regarding the Weakness #1: Regarding the first question, the three primary error causes were identified through our error analysis. We have conducted additional experimental analyses for each of these causes, detailed in their respective subsections. For the 'tractor' example in Figure 2, we will include descriptions for each category in the task instructions in future experiments. This will help us more accurately differentiate between 'unspecific context understanding' and 'Misaligned Schema Understanding' as reasons of error.
>
> For the second question, the prompts for different models consist of instructions (as provided in Appendix A.1) and 8 shots of demonstrations. The term 'specification' specifically refers to these instructions, which vary in length according to the different instructions required for each task. Regarding the reasons for performance decline, I appreciate this insightful query. We will conduct additional experiments, such as tasks with long context but easy specifications, to ascertain whether the decline in performance is due to the long context or heavy specifications.
>
> Regarding the Weakness #2: The definition of 'specification-heavy' tasks we propose refers to tasks with complicated and extensive task specifications, which would typically require several hours for an ordinary person to learn and perform competently, though not necessarily perfectly. For such tasks, significant learning time is essential to achieve a decent level of proficiency, not necessarily to solve the task flawlessly.
>
> In response to the observation that 'most NLP tasks involve “complex” annotation guidelines for detailed task descriptions and edge cases,' it's important to note that many tasks, like the NLI task, can be performed satisfactorily in a relatively short time. The 'detailed task descriptions and edge cases' are requirements for perfect execution, which does not categorize these as specification-heavy tasks. Thank you for your valuable suggestion. We will incorporate additional tasks and provide clearer and more intuitive descriptions for them in our revision.
>
> Regarding the Weakness #3: Thank you for your suggestion. We have responded to this issue in our general response. We will conduct additional experiments using detailed prompts.
>
> Regarding the Question #1: Thank you for your valuable suggestion. In line with your advice, we will provide further analysis and insights based on the results of our experiments.
>
> Regarding the Question #2: Thank you for your valuable suggestion. We will conduct a similar error analysis on instruction-tuned models to observe the proportion of errors corrected by instruction tuning.
>
> Regarding the Question #3: Thank you for the reminder. We will cite it in our subsequent revisions.
>
> Regarding the Question #4: Thank you for your detailed review. We will thoroughly review and correct the grammatical errors and typos in the paper.
>
> Thank you for your valuable suggestions. We are committed to supplementing our research with further experiments.

---

> > ### Comment · Reviewer_Huq3 · 2023-11-23
> >
> > Thanks for answering my questions. Looking forward to see your update.

---

### Official Review · Reviewer_KipK · 2023-10-31

**Soundness:** 3 good
**Presentation:** 3 good
**Contribution:** 3 good
**Rating:** 5
**Confidence:** 4

**Summary:**

The paper studies using the in-context-learning (ICL) paradigm with LLMs for solving specifications-heavy tasks, i.e., tasks that need to be described with a lengthy set of instructions such as relation extraction. Using 18 benchmarks (6 different task types) and six LLMs (FLAN-UL2, Alpaca, Vicuna, ChatGPT, Davinci003 and GPT-4), the paper demonstrates that few-shot ICL significantly lags behind the SOTA of each task. Manual analysis highlights three reasons for that poor performance: 1) inability of the ICL+LLM approach to understand specifics of the context, 2) lack of task schema comprehension, and 3) limited LLM ability to understand long context. The paper then confirms that the poor performance is due to the ICL approach itself rather than the capabilities of the LLM by fine-tuning FLAN-UL2 specifically for each task and showing accuracies that outperform SOTA. The paper argues that the current LLM alignment datasets do not well cover complicated specification-heavy tasks. Via supervised fine-tuning, the paper aligns FLAN-UL2 using the corresponding training  data of each task (leaving one task out for evaluation) and shows gains in accuracy compared to that obtained before alignment.

**Strengths:**

1. The paper highlights an interesting challenge to the in-context-learning paradigm that even today's most powerful LLMs struggle with.

2. The paper presents promising initial results that demonstrates the potential of addressing the challenge of specification-heavy tasks via more focused alignment.

**Weaknesses:**

1. The success of the ICL approach highly depends on the amount and quality of information provided in the prompts. The results in table 1 are all based on a shortened version of the task description. Examples in the appendix show that such shortened descriptions are too concise, e.g., they do not even contain a single sentence definition of each label. The provided few-shots do not cover all labels. Longer prompts can still fit in at least a subset of the LLMs used for the experiments. The paper needs to experiment with more detailed prompts to confirm that the poor accuracy of the ICL approach is not due to the very concise set of instructions.

2. The alignment experiments in Section 4.2 only hold out one task at a time, but still tasks of the same type are included in the alignment dataset. The paper needs to report results with a whole task type (e.g., All 4 RE tasks) is held out to confirm that the gain in accuracy is not really due to similarity between tasks.

**Questions:**

1. Comparing table 1 to table 4, Aligned ICL has worse accuracy than the unaligned  baseline on 3 tasks MAVEN-Subevent and the two sentiment analysis tasks. Do you have an explanation or an intuition for why that is the case for those three datasets?

2. In Section 3.1, the paper says that "LLMs ignore all the contexts in 18 instances". How did you find out about that?

---

> ### Author Response · Authors · 2023-11-21
>
> Thank you for your valuable review!
>
> Regarding the Weakness #1: Thank you for your suggestion. We respond to this issue in our general response. We will conduct additional experiments using more detailed prompts.
>
> Regarding the Weakness #2: Thank you for your valuable suggestion. Holding out a task each time is a more reasonable approach. In light of this, we plan to revise the settings of all our experiments to hold out one task, in line with the practices in the FLAN series.
>
> Regarding the Question #1: The inferior performance on the MAVEN-Subevent task may be attributed to the sparse annotations and complex output requirements of this type of task, which could impede the effective generalization of Aligned ICL. As for the diminished performance on the two sentiment analysis tasks, this could be due to the inclusion of sentiment analysis tasks in the original alignment corpus of FLAN-UL2. Consequently, FLAN-UL2 might perform well on these tasks even without our alignment. However, the relatively low scale of sentiment classification in our used alignment data could lead to reduced performance of the FLAN-UL2 aligned on our data in sentiment analysis tasks.
>
> Regarding the Question #2: This conclusion is drawn from our comparative experiments. When we removed all context and merely posed the corresponding questions to GPT-4, the model was still able to correctly answer these questions. From this observation, we infer that the LLM does not utilize the context information, which we term as 'ignoring all the context.’

---

> > ### Comment · Reviewer_KipK · 2023-11-22
> > **Thanks**
> >
> > Thanks for answering my questions and for acknowledging the highlighted directions for future improvements.

---

### Official Review · Reviewer_3cva · 2023-11-10

**Soundness:** 2 fair
**Presentation:** 2 fair
**Contribution:** 1 poor
**Rating:** 3
**Confidence:** 4

**Summary:**

The paper discusses the limitations of in-context learning (ICL) in dealing with tasks that have complex and detailed specifications.
It particularly highlights the challenges ICL faces in information extraction tasks like Named Entity Recognition (NER) and Relation Extraction (RE), which it categorizes as specification-heavy.
The authors attribute ICL's shortcomings in these areas to three primary factors:

1. A lack of precise comprehension of the context.
2. Misinterpretation of the task schema.
3. Ineffectiveness in processing and understanding long texts.

**Strengths:**

The paper illustrates that instruction-following language models struggle to effectively utilize demonstration examples provided in their instructions, particularly when dealing with tasks that have heavy specification requirements.

**Weaknesses:**

I have a few major concerns:

1. Clarification of the definition of in-context learning.

In-context learning (ICL) is typically associated with a few-shot learning scenario that doesn't include explicit task instructions [1,2,3]. However, the ICL prompts discussed in the paper are characterized by a combination of 'Instruction + Demonstrations', diverging from the conventional ICL format, which usually involves only 'Demonstrations'.

2. Performance of `instruction + demonstrations`

The effectiveness of prompts that include both 'instruction and demonstrations' might be enhanced if large language models (LLMs) are specifically trained with such prompts. This training approach could potentially improve their performance on these tasks.

3. Labels covered in demonstrations are important.

It's crucial to show how demonstrations incorporate all labels in the label space.
Typically, k-shot learning implies presenting k examples for each label.
If the demonstrations are limited due to the constraints of input context length, this could understandably lead to lower performance.

*[1] Larger language models do in-context learning differently., Wei et al., 2023 \
[2] Rethinking the role of demonstrations: What makes in-context learning work?., Min et al., 2023 \
[3] Large Language Models as General Pattern Machines., Mirchandani et al., 2023*

**Questions:**

Same as weakness section.

---

> ### Author Response · Authors · 2023-11-21
>
> Thanks for your review!
>
> For the Weakness #1: We politely disagree that in-context learning can only be performed with demonstrations. The original definition of ICL in the GPT-3 paper [1] is explicitly characterized as encompassing both instructions and demonstrations. Additionally, if ICL were to include only demonstrations, the community would have no rationale to create a new term instead of directly employing the widely-accepted term 'few-shot learning'.
>
> For the Weakness #2: It appears there may be some ambiguity in the question posed. We have employed instruction tuning method to enhance the LLM's ability to follow instructions. This approach has not only improved the model's performance across multiple tasks but also on tasks that it has not previously encountered. I hope this clarification addresses your query effectively.
>
> For the Weakness #3: Thank you for your valuable suggestion. Firstly, our experimental setup adheres to the conventional usage established in previous research [2,3], namely employing a certain number (8, 16, or, 32) of 'shot' examples for in-context learning. Indeed, the exclusion of some labels from the label space in our demonstrations may affect the model's effectiveness. We appreciate this insight and will incorporate this aspect in future experiments with models that can accommodate longer inputs.
>
> [1] Brown, Tom B. et al. “Language Models are Few-Shot Learners.” *ArXiv* abs/2005.14165 (2020): n. pag. https://arxiv.org/pdf/2005.14165.pdf
>
> [2] Han, Ridong et al. “Is Information Extraction Solved by ChatGPT? An Analysis of Performance, Evaluation Criteria, Robustness and Errors.” *ArXiv* abs/2305.14450 (2023): n. pag.
>
> [3] Yu, Jifan et al. “KoLA: Carefully Benchmarking World Knowledge of Large Language Models.” *ArXiv* abs/2306.09296 (2023): n. pag.

---

### Author Response · Authors · 2023-11-21

Thank you for your meticulous reviews, which are invaluable for helping us improve this paper. We have summarized the common concerns raised and provide our responses here.

A major concern among reviewers is the insufficiency of the prompts, such as the lack of descriptions for classes and the failure to cover the entire label space in the demonstrations, leading to suboptimal performance of LLMs on these tasks. In Table 3 and 16 of the paper, we conducted experiments with descriptions of classes. These experiments indicate that even with the descriptions, the improvement in LLM performance is limited and may even lead to a decrease. Regarding the incomplete coverage of the label space, our approach was to follow the common practice of previous works [1,2], which sample a number of examples as demonstrations. This indeed might contribute to reduced effectiveness and we will try to mitigate this in our future version.

We appreciate the valuable suggestions from reviewers. In our revised version, we will employ more detailed prompts, such as more detailed instructions and more demonstrations to enhance the credibility of our results.

[1] Han, Ridong et al. “Is Information Extraction Solved by ChatGPT? An Analysis of Performance, Evaluation Criteria, Robustness and Errors.” *ArXiv* abs/2305.14450 (2023).

[2] Yu, Jifan et al. “KoLA: Carefully Benchmarking World Knowledge of Large Language Models.” *ArXiv* abs/2306.09296 (2023).

---

### Meta-Review · Area_Chair_nedR · 2023-12-13

**Metareview:**

The paper investigates the application of the in-context-learning (ICL) in Large Language Models (LLMs) for specifications-heavy tasks, such as relation extraction. It finds that few-shot ICL faces challenges in understanding context specifics, limited comprehension of task schema, and the LLM's constrained ability to grasp long contexts. Reviewers raised concerns regarding the experimental set up and lacking of enough analysis.

**Justification For Why Not Higher Score:**

NA

**Justification For Why Not Lower Score:**

NA

---

### Decision · Program_Chairs · 2024-01-16

Reject